# Influence of Sand-Cement Ratio and Polycarboxylate Superplasticizer on the Basic Properties of Mortar Based on Water Film Thickness

**DOI:** 10.3390/ma14174850

**Published:** 2021-08-26

**Authors:** Zhao Zhang, Qingge Feng, Weiwei Zhu, Xianhao Lin, Kao Chen, Wuxiao Yin, Changhai Lu

**Affiliations:** 1School of Resources, Environment and Materials, Guangxi University, Nanning 530004, China; z1141466379@163.com (Z.Z.); guet_lin@163.com (X.L.); 2Materials and Environmental Engineering, Guangxi University for Nationalities, Nanning 530004, China; zhuww1230@163.com; 3School of Chemistry and Chemical Engineering, Guangxi University, Nanning 530004, China; chen1kao@163.com; 4Guangxi Construction Testing Centre, Nanning 530004, China; yinwuxiao2010@163.com (W.Y.); 18176891861@163.com (C.L.)

**Keywords:** sand-cement ratio, PCE, mortar, water film thickness

## Abstract

Previous studies demonstrated that water film thickness (WFT) is a key factor that affects the fluidity of mortar. Changes in the sand-cement (S/C) ratio and polycarboxylate superplasticizer (PCE) dosage will affect the WFT. In this study, several mortar samples with different S/C ratios and different PCE dosages were prepared, and the basic properties of the mortar were measured. The results show that as the S/C ratio increases, the packing density of the mortar will decrease, the WFT will decrease, and the cohesiveness will increase, resulting in a decrease in the flow spread and strength of the mortar. When the PCE dosage is increased, the packing density of the mortar will increase, the WFT will increase, and the cohesiveness will decrease, which increases the flow spread of the mortar. When the water-cement (W/C) ratio is low, the S/C ratio has a significant effect on the strength, and the strength will increase with the increasing of the PCE dosage. When the W/C ratio is high, the strength of the mortar will be reduced once the PCE dosage exceeds the saturation value. In the case of different S/C ratios or different PCE dosages, the WFT can be used as a measure of mortar cohesiveness and flow spread.

## 1. Introduction

In recent decades, water film thickness (WFT) was extensively studied as an important factor that affects the cohesiveness and fluidity of paste, mortar, and concrete [1,2]. WFT is defined as the average thickness of water wrapped on the surface of solid particles [3]. As the WFT increases, it can be predicted that the flow spread of cement paste will increase [4,5]. This is because the increase in water on the surface of solid particles reduces the friction between particles [6]. Li, Kwan, and others conducted extensive research in this regard [7,8]. The calculation principle of the WFT is obtained by dividing the excess water that remains after filling the voids of cement particles by the specific surface area of the material [9,10]. It is indispensable to determine the packing density of materials to obtain the excess water volume, which is used to calculate the WFT [11]. There are two methods for measuring the packing density of materials. One is the conventional measurement method under dry conditions. However, this method is not applicable to mortar containing fine aggregates and cementitious materials, as fine aggregate and cementitious materials agglomerate easily under dry conditions, significantly influencing the measurement results [12]. On the other hand, when measured under dry conditions, additives related to cement cannot be used, and the effect of additives cannot be reflected. The second method was developed by Kwan et al., and it determines the packing density of materials under wet conditions. This method avoids the two shortcomings of dry packing, and therefore meets the experimental requirements of this research [13].

In the existing research on the influence of the WFT on mortar, the sand–cement (S/C) ratio is an influencing factor that is rarely accounted for. However, sand plays the role of skeleton and connection in the mortar; it also plays important roles in the internal structure and mechanical properties [14]. Secondly, the amount of sand will affect other aspects such as application costs. To make mortar with good flow spread, more water needs to be added with the S/C ratio increases [15]. Changes in the S/C ratio will affect the surface area of materials [16] and the packing density of the mortar, leading to changes in the WFT and affecting mortar performance [17,18].

As one of the most important additives to cement, the polycarboxylate superplasticizer (PCE) can allow mortar to have better flow spread at a low W/C ratio [19]. This is because the PCE can be adsorbed on the surface of cement particles and possesses the same electric charge [20,21]. The mutual repulsion of the same type of electric charge leads to the dispersion of cement particles [22]. On the other hand, the side chains of PCE are relatively long and stretch in the slurry [23], thus increasing the steric repulsion between the cement particles and improving the dispersion effect [24]. The flocculation structure of the cement paste is opened, increasing the content of free water in the paste and improving the fluidity [25]. Previous studies showed that the material packing density will be higher with a wider range of material particle sizes [26]. The PCE can disperse cement particles and thus increase the packing density of cement paste [27], leading to an increase in the WFT [28,29] and further improving the flow spread of the paste. However, if the PCE dosage is too high, it will cause the cement paste to bleed and affect other properties such as cohesiveness and strength.

Previous studies showed that when the W/C ratio is high, the addition of PCE is likely to cause bleeding of the mortar [30]. Furthermore, the presence of the proper S/C ratio and PCE dosage are critical for mortar because the S/C ratio affects the optimal value of the W/C ratio. The WFT is also a key factor to the fluidity as well as other properties of the mortar. Different S/C ratios and PCE dosages can change the WFT and thus affect the performance of the mortar. Although previous studies were carried out to investigate the effects of PEC dosages on the WFT, for these investigations, it was difficult to explain the impact of the S/C ratios on the WFT [31]. However, the PCE dosages and the S/C ratios are indispensable factors in the actual application of mortar. To resolve the issues explained above, some mortar samples were prepared with different S/C ratios and PCE dosages, and the packing density, cohesiveness, flow spread, and strength were measured. According to the experimental results, the cohesiveness and flow spread of the mortar samples were regression analyzed with the WFT. From this point to in-depth analysis the effects of S/C ratios and PCE dosages on the basic properties of the mortar by changing the WFT.

## 2. Materials and Methods

### 2.1. Materials

The reference cement (RC) used in the experiment was produced in Beijing, and the fine aggregate was Chinese ISO standard sand produced in Xiamen, with a maximum particle size of 1.10 mm. Table 1 shows the chemical composition and physical properties of RC and fine aggregate. The particle size distributions of RC and fine aggregate were determined using the Mastersizer 3000 laser diffraction particle size analyzer that is made in England, and they were shown in Figure 1. PCE was provided by Jiangsu Subote Chem. Reagent Company and characterized with a solid content of 20%, a water reduction rate of 21%, and a density of 1030 kg/m^3^.

### 2.2. Mix Design

To study the effect of the S/C ratio and the PCE dosage on the basic properties of mortar, the experiment adopted five kinds of S/C ratio and five kinds of PCE dosage. Based on Chinese standard JGJ/T 98-2010, JGJ/T220-2010, and JG/T 223-2007, and to avoid bleeding of the mortar in the experiment, the S/C ratios we designed (by mass) were 2.4, 2.6, 2.8, 3.0 and 3.2, respectively; the PCE dosages (by mass) accounted for 0%, 0.1%, 0.2%, 0.3% and 0.4% of the reference cement; and the W/C ratios (by mass) were 0.40, 0.45, 0.50, 0.55 and 0.60, respectively. The mixture ratios of the mortar samples are shown in Table 2.

For easy identification, X-Y-Z is specified for each mortar sample, where X represents the S/C ratio, Y represents the PCE dosage, and Z represents the W/C ratio. For example, 2.4-0.2-0.50 indicates a mortar sample with an S/C ratio of 2.4, a PCE dosage of 0.2%, and a W/C ratio of 0.5. For cohesiveness, flow spread, and strength of tests, each experiment was repeated three times and the results were averaged.

### 2.3. Testing Methods

#### 2.3.1. Measuring Packing Density and Calculating Water Film Thickness

The packing density of the mortar adopts the wet packing test [32]. The packing density appears when the amount of water is just sufficient to fill the gaps between the materials and form a wet cement paste. This is when the particle spacing is at its smallest, and the bonding is at its tightest. In the experiment, the W/C ratio was changed from high to low, the prepared cement paste was poured into a 60 mm × 58 mm cylindrical mold, and the total mass of the mold and cement paste was weighed with a balance. The packing density was calculated using the following equation [33].
(1)$$P=\frac{M/V}{R_{c}\rho_{c}+R_{s}\rho_{s}+v_{w}\rho_{w}}$$
where *P* represents the packing density; *V* represents the volume of the mold. *M* is the mass of the cement mortar paste; *R_c_* and *R_s_* are the volume fractions of cement and fine aggregate in the total material, respectively; and *v_w_* is the volume fraction of water in the total material. *ρ_c_, ρ_s_, ρ_w_* are the densities of cement, fine aggregate, and water, respectively.

When the amount of water added to the mortar was sufficient to fill the voids of the mortar, the excess water would wrap on the surface of the material to form a water film [34]. The WFT was calculated using the following equations [35].
(2)$$T=\frac{W_{e}}{A}$$
(3)$$W_{e}=V_{w}-V_{p}$$
(4)$$V_{p}=(\frac{1}{P_{\max}}-1)(M_{C}/\rho_{C}+M_{S}/\rho_{S})$$
(5)$$A=M_{c}A_{c}+M_{s}A_{s}$$
where *T* is the thickness of the water film; *W_e_* is the excess water volume; *A* is the total surface area of cement and fine aggregate; *V_w_* is the actual water volume of the sample; *V_p_* is the void volume between particles; *M_c_* and *M_s_* are the actual masses of cement and fine aggregate used in the sample, respectively; *ρ_c_* and *ρ_s_* are the densities of cement and fine aggregate, respectively; *P_max_* is the packing density of mortar; *A_c_* and *A_s_* are the specific surface areas of the cement and fine aggregate, respectively.

#### 2.3.2. Measuring Cohesiveness

The cohesiveness of the mortar sample was characterized by the ratio of the weight of the mortar passing through the sieve to the total mass. For the test, approximately 500 g of mortar was placed to stand for 15 min and then poured onto a 4.75 mm sieve from a height of 300 mm. After two minutes, the mortar would drip through the sieve and then be collected by the base receiver before being weighed. The proportion of mortar dripping through the screen could be used as a measure of cohesiveness. The more mass that passed through the sieve, the less cohesiveness it was. After the test, the sieve segregation index (SSI) of the tested mortar sample could be determined [36].
(6)$$SSI=\frac{M_{p}}{M_{m}}\times 100\%$$
where *M*_p_ is the mass of mortar collected by the base receiver, and *M*_m_ is the mass of mortar poured on the sieve.

#### 2.3.3. Measuring Flow Spread

The flow spread of mortar was measured using the current “Method for Measuring the Fluidity of Cement Mortar Sand” (GB/T2419-2005). In accordance with the procedure in the standard, the prepared mortar was poured into a truncated cone. After completion, the truncated cone round mold was slowly lifted vertically upwards and the measuring instrument (jumping table) was started, and 25 beatings were completed at a frequency of 1 time per second. The average of the two vertical diameters of the mortar was measured until the vibration was completed. Then, the base diameter of the truncated cone mold was subtracted from the calculated average diameter of the mortar. The final value was the flow spread of the mortar [37].

#### 2.3.4. Measuring Strength

The strength of mortar was measured using the current “Standard for Test Method of Performance Building Mortar” (JGJ/T70—2009). The stirred mortar was poured into a cube mold with a side length of 70.7 mm for 5 s vibration, and six samples were prepared. The mold was removed after 24 h, and the mortar samples were cured indoors at (20 ± 3) ℃ and humidity above 90% for 7 d and 28 d. The results of compressive strength were obtained from the average value of three mortar specimens [38].

## 3. Results and Discussion

### 3.1. Packing Density

The results of the packing density of the solid particles in the mortar are shown in Table 3. To illustrate how the packing density changes with the S/C ratio, the packing densities for five S/C ratios were plotted as a curve. As shown in Figure 2, the packing density of the mortar decreases with the increasing of the S/C ratio. During the packing process, the particle size of fine aggregate is much larger than that of the cement particles. The increase in the S/C ratio will increase the voids, causing the mortar to be less dense and the packing density to decrease [39].

To illustrate how the packing density changes with the PCE dosage, the packing densities for five different PCE dosages were plotted as a curve. Figure 3 indicates that the packing density of the mortar increases with the PCE dosage. During the packing process, PCE disperses the cement particles, making the mortar more compact and thus increasing the packing density.

### 3.2. WFT Results

The results of water film thickness of mortar samples are shown in Table 3. When the WFT value is positive, it means the average thickness of water coated on solid particles. However, when the WFT value is negative, the WFT does not have such a significance. A negative WFT value indicates that the added water is not sufficient to fill the voids in the mortar. To explain the impact of S/C ratio variations on WFT, the correlation between the WFT and W/C ratio for five S/C ratios was plotted as a scatter curve, and then linear fitting was performed. As can be seen in Figure 4a, when the S/C ratio increases, WFT decreases. The packing density of the material will decrease and its surface area will increase under the condition of a rising S/C ratio. These will result in a decrease in WFT. The R2 value is, in all cases, 0.999, and the regressed parameters are shown in Table 4. According to the fitting formula, the W/C ratio (critical value) can be calculated when the water film has just formed, as illustrated in Table 4. The critical value will increase under the condition of a rising S/C ratio. Mainly due to the increased surface area of the material, more water needs to be added to form a water film.

Similarly, to illustrate how the WFT changes with the PCE dosage, the correlation between the WFT and W/C ratio was plotted as a scatter curve for five different PCE dosages, and then linear fitting was performed. This is illustrated in Figure 4b. The R2 value is, in all cases, 0.999. When the PCE dosage increases, WFT increases. The increase in PCE dosage increases the packing density of the mortar and leads to an increase in WFT. As shown in Table 4, the critical value will decrease with the increasing of the PCE dosage. The PCE can disperse cement particles and open up the flocculation structure to increase the amount of free water. Thus, the critical value is lowered.

### 3.3. Cohesiveness

To explain how the cohesiveness varies with the S/C ratio, the SSI results and the W/C ratios under five S/C ratios were plotted as a curve, as illustrated in Figure 5a. The SSI increases with the increase in the W/C ratio under the condition of a constant S/C ratio. The increased W/C ratio will damage the cohesiveness of the mortar. Comparing the different curves under different S/C ratios, it can be seen that with an W/C ratio greater than 2.8, the SSI rises slowly with the increasing of the W/C ratio. On the contrary, the rising speed of SSI increases significantly. The speed of SSI change depends on the S/C ratio. Under the condition of a W/C ratio of 0.40, the SSI of the mortar with the S/C ratio of 3.2 is 0%, the SSI of the mortar with the S/C ratio of 2.4 is 1.03%, and the SSI of the mortar increases by 1.03%. Under the condition of a W/C ratio of 0.60, the SSI of the mortar with the S/C ratio of 3.2 is 1.77%, while the SSI of the mortar with the S/C ratio of 2.4 is 13.97%, and the SSI of the mortar increases by 13.97%. This shows that when the W/C ratio is high, the S/C ratio can clearly affect the cohesiveness of the mortar.

The correlation between the SSI and W/C ratio under five PCE dosages, plotted as a curve, illustrates how the PCE dosage affects cohesiveness. As can be seen in Figure 5b, when the PCE is not mixed or the PCE dosage is low, the W/C ratio increases, and the SSI increases slowly. When PCE dosage is higher than 0.3%, the rising speed of SSI increases significantly. It can be seen that the speed of the SSI increase depends on the PCE dosage. Under the condition of a W/C ratio of 0.40, the SSI of the mortar without PCE is 0%, the SSI of the mortar with a PCE dosage of 0.4% is 0.68%, and the SSI of the mortar increases by 0.68%. Under the condition of a W/C ratio of 0.60, the SSI of the mortar without PCE is 1.81%, the SSI of the mortar with a PCE dosage of 0.4% is 9.99%, and the SSI of the mortar increases by 8.18%. This shows that when the W/C ratio is high, the influence of PCE dosage on the cohesiveness of the mortar is more obvious.

### 3.4. Flow Spread

To explain how the S/C ratio changes with the flow spread, the curves of the flow spread against the W/C ratio, under five different S/C ratios, were plotted. As illustrated in Figure 6a, the flow spread increases with the increasing of the W/C ratio under the condition of a constant S/C ratio. In the comparison of the different curves under five S/C ratios, it can be observed that the increase in flow spread with the decreasing of the S/C ratio is highly apparent. More water is needed to keep the flow spread from decreasing when the S/C ratio increases. Under the condition of a W/C ratio of 0.40, the flow spread of the mortar with the S/C ratio of 3.2 is 0 mm, the flow spread of the mortar with the S/C ratio of 2.4 is 53.5 mm, and the SSI of the mortar increases by 53.5 mm. Under the condition of a W/C ratio of 0.60, the flow spread of the mortar with the S/C ratio of 3.2 is 82.5 mm, the flow spread of the mortar with the S/C ratio of 2.4 is 200 mm, and the SSI of the mortar increases by 117.5 mm. It can be seen that when the W/C ratio is high, the S/C ratio has a more obvious effect on the flow spread of the mortar.

To illustrate the change in flow spread with different PCE dosages, the flow spreads and W/C ratios, under five different PCE dosages, are plotted as a curve [40]. This is illustrated in Figure 6b. By comparing different curves at different doses of PCE, it can be observed that the flow spread does not change significantly for different PCE dosages at a 0.40 W/C ratio. Without PCE, the flow spread of the mortar will be 0 mm when the W/C ratios are 0.4 and 0.45. Only when the W/C ratio increases can it have a certain degree of fluidity. When the SP dosage is 0.1% and the W/C ratio increases to above 0.50, the mortar starts to flow. When the PCE dosage is 0.2% and the W/C ratio is 0.40, the mortar starts to flow. Therefore, the addition of PCE can make the mortar flow at a lower W/C ratio. Under the condition of a W/C ratio of 0.40, the flow spread of the mortar without PCE is 0 mm, the flow spread of the mortar with a PCE dosage of 0.4% is 23 mm, and the flow spread of the mortar increases by 23 mm. Under the condition of a W/C ratio of 0.60, the flow spread of the mortar without PCE is 76 mm, the flow spread of the mortar with a PCE dosage of 0.4% is 143 mm, and the flow spread of the mortar increases by 67 mm. It is obvious that when the W/C ratio is high, the influence of PCE dosage on the flow spread of the mortar is significant.

### 3.5. Strength

To explain how the S/C ratio affects strength, the strength and W/C ratios under five S/C ratios were plotted as a curve. As illustrated in Figure 7, under the condition of a constant S/C ratio, the strength will decline with the increasing of the W/C ratio. Comparing the different curves under five S/C ratios, the strengths of samples are similar at a 0.6 W/C ratio, indicating that the S/C ratio has no obvious effect on the strength under the condition of a high W/C ratio. In contrast, when the W/C ratio is lower than 0.50, an increased S/C ratio will significantly reduce the strength of the mortar specimen. This shows that the influence of the S/C ratio on the strength is more obvious under the condition of a low W/C ratio. Cement is the only gel material. In the same volume of mortar specimen, the S/C ratio increases, and the cement content of the mortar specimens is relatively less. Secondly, an increase in the S/C ratio increases the gaps between particles, resulting in lower strength.

To explain how the PCE dosage affects strength, the correlation between the strength and PCE dosage for five different W/C ratios was plotted as a curve. This is illustrated in Figure 8. Under the condition of 7 days of curing of the mortar specimens, when the PCE dosage is unchanged, the strength decreases with the increasing of the W/C ratio. In the comparison of curves with different W/C ratios, when the W/C ratio is less than 0.5, the strength will increase with the increasing of the PCE dosage. This is due to the dispersion of the PCE, which makes the mortar more compact. Secondly, the cement particles are fully in contact with the water, and the hydration reaction is more complete. At the W/C ratio of 0.55, when the PCE dosage is greater than 0.3%, the strength will decrease to a certain extent. Furthermore, at the W/C ratio of 0.60, when the PCE dosage is greater than 0.2%, the strength will also decrease to a certain extent. This indicates that when the W/C ratio is high and the PCE dosage has a saturation value, the strength will be reduced. Regardless of the S/C ratios and PCE dosages, the mortar specimens cured for 28 days demonstrated higher strengths than those cured for 7 days. The 28 day period of curing allows the hydration to be more complete and results in higher strength.

## 4. Roles of Water Film Thickness

### 4.1. Effects of WFT on Cohesiveness

To study the relationship between WFT and cohesiveness under different S/C ratios, the SSI corresponding to WFT under five different S/C ratios was plotted as a scatter plot. Then, nonlinear fitting was performed, as shown in Figure 9a. The R^2^ value is 0.985, which shows that WFT and cohesiveness have a good correlation, and that cohesiveness will decrease with the increasing of the WFT. As the WFT increases, the distance between the particles becomes larger, the gravitational force decreases [41], and the mortar cohesiveness decreases [42]. A decrease in the S/C ratio will increase the WFT and reduce the cohesiveness of the mortar. It can also be seen from Figure 9a that when the WFT is negative, SSI is less than 1%. This is because the added water is not sufficient to fill the voids of the mortar and the distance between the particles is very close. When the WFT is positive, the increased rate of SSI gradually increases as the S/C ratio decreases. The S/C ratio has a more obvious effect on the cohesiveness of the mortar. This is because when the WFT is positive, the decrease in the S/C ratio will increase the distance between particles.

To explain the relationship between WFT and cohesiveness under different PCE dosages, the SSI corresponding to the WFT under five different PCE dosages was plotted as a scatter diagram, and then nonlinear fitting was performed. This is illustrated in Figure 9b. The R^2^ value is 0.949, which also shows that WFT and cohesiveness have a good correlation. This indicates that cohesiveness will decrease with the increasing of the WFT. The dispersing effect of the PCE can disperse the flocculated cement particles and damage the cohesiveness of the mortar. Both the WFT and PCE dosage significantly influence the cohesiveness. It can also be seen from Figure 9b that when the WFT is negative, the SSI is also less than 1%. This is because a layer of water film cannot be formed on the surface of the cement particles, and a small amount of PCE acts on the surface of the cement particles. When WFT is positive, the increased rate of SSI gradually increases under the condition of rising PCE dosage. This is because when WFT is positive, more PCE acts on the surface of cement particles to have a more obvious effect on the cohesiveness of the mortar. Regardless of any changes in the S/C ratio or PCE dosage, the relationship between the SSI and WFT for every mortar sample can be described by the following equation. This illustrates that the cohesiveness of mortar can be determined by calculating the WFT.
(7)$$FS=a*(1+WFT{)}^{b}$$

### 4.2. Effects of WFT on Flow Spread

To study the relationship between WFT and flow spread under different S/C ratios, the flow spread corresponding to WFT under five different S/C ratios was plotted as a scatter plot, and then nonlinear fitting was performed, as illustrated in Figure 10a. The R^2^ value is 0.986. This shows that WFT and flow spread has a good correlation, and the flow spread of mortar increases with the increasing of the WFT. The increase in the WFT reduces the friction between the solid particles, so the flow spread of the mortar will increase. The S/C ratio changes the flow spread of the mortar by influencing the WFT. When the WFT is negative, the flow spread is less than 50 mm. This is due to the low free water content in the mortar, which leads to the low flow spread of the mortar. When the WFT is positive, the increased rate of mortar flow spread will increase to a certain extent as the S/C ratio decreases. This is because when the WFT is positive, the decrease in the S/C ratio leads to an increase in the WFT, which reduces the friction between the particles and has a better effect on the flow spread of the mortar.

To explain the relationship between WFT and flow spread under different PCE dosages, the flow spread corresponding to the water film thickness under five different PCE dosages was plotted as a scatter plot, and then nonlinear fitting was performed, as shown in Figure 10b. The R^2^ value is 0.972, which also shows that WFT and flow spread have a good correlation. When the WFT increases, the flow spread of the mortar increases. The dispersing effect of PCE makes it easier for the mortar to flow. Both the WFT and PCE dosage influence the flow spread of the mortar. It can also be seen from Figure 10b that when the WFT is negative, the flow spread is also less than 50 mm. This is because a layer of water film cannot be formed on the surface of the cement particles, which results in a low effect of PCE. When the WFT is positive, the increased rate of mortar flow spread will increase to a certain extent as the PCE dosage increases. This is because when the WFT is positive, the PCE has a more obvious effect on the flow spread of the mortar. Whether the S/C ratio or the PCE dosage changes, the relationship between the flow spread (FS) and the WFT for every mortar sample can be described by the following equation. This illustrates that the flow spread of mortar can be determined by calculating the WFT.
(8)$$FS=a*(WFT-b{)}^{c}$$

## 5. Conclusions

An increase in the S/C ratio will reduce the packing density of the mortar and the WFT but will also increase the cohesiveness, resulting in a decrease in the flow spread and mortar strength. The increase in the PCE dosage will increase the packing density of the mortar and the WFT but decrease the cohesiveness, leading to an increase in the flow spread of the mortar. When the W/C ratio is low, the S/C ratio has an obvious effect on strength, and the strength will increase with the increasing of the PCE dosage. When the W/C ratio is high, the PCE dosage has a saturation value. The PCE dosage exceeds the saturation value, and the strength of the mortar decreases as the PCE dosage increases.Different S/C ratios and PCE dosages affect the WFT, resulting in changes in the cohesiveness and flow spread of the mortar. When the WFT is positive, the effects of the S/C ratio and PCE dosage on the cohesiveness and flow spread of the mortar are more obvious. When the value of WFT is positive, the mixing ratio of the mortar is acceptable.Regardless of any changes in the S/C ratio or PCE dosage, the WFT has a good correlation with the flow spread and mortar cohesiveness. The WFT was calculated by the packing density of the mortar, which determined the cohesiveness and flow spread of the mortar. For every mortar sample, the relationship between the SSI and the WFT can be described by SSI = a × (1 + WFT)^b, and the relationship between the flow spread and the WFT can be described by FS = a × (WFT-b)^c. The cohesiveness and flow spread of the mortar can be predicted by the WFT.

## Figures and Tables

**Figure 1 materials-14-04850-f001:**
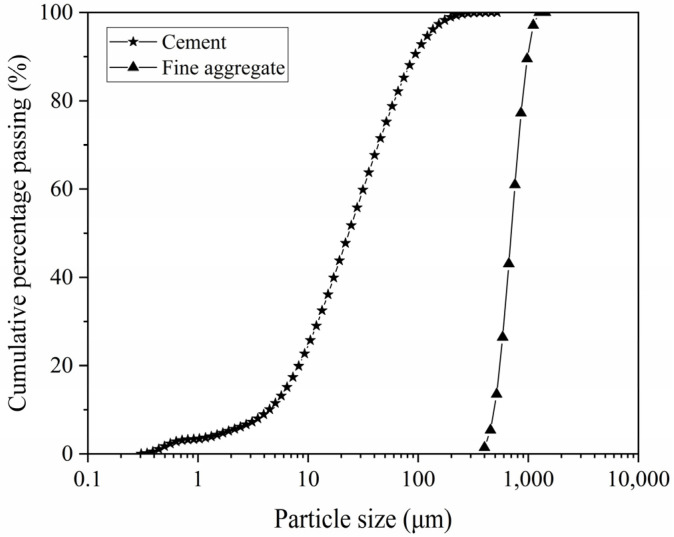
Cumulative particle size distributions of RC and fine aggregate.

**Figure 2 materials-14-04850-f002:**
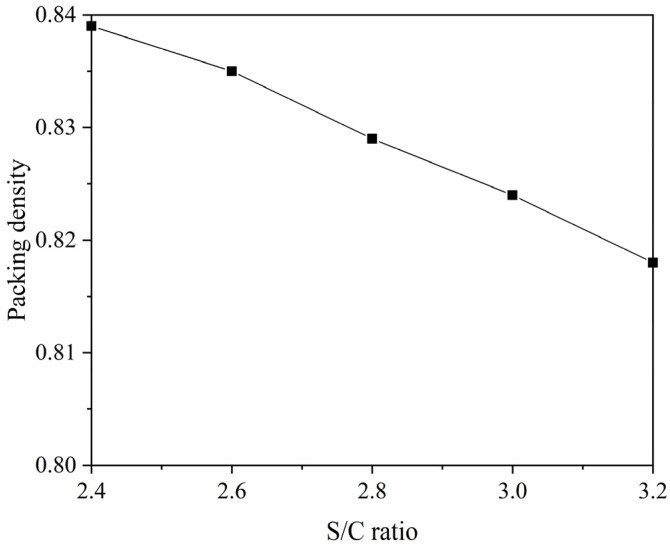
Packing density versus S/C ratio.

**Figure 3 materials-14-04850-f003:**
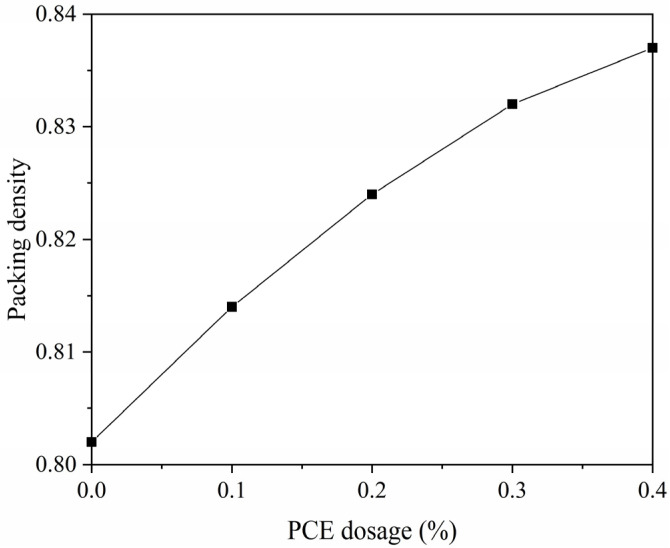
Packing density versus PCE dosage.

**Figure 4 materials-14-04850-f004:**
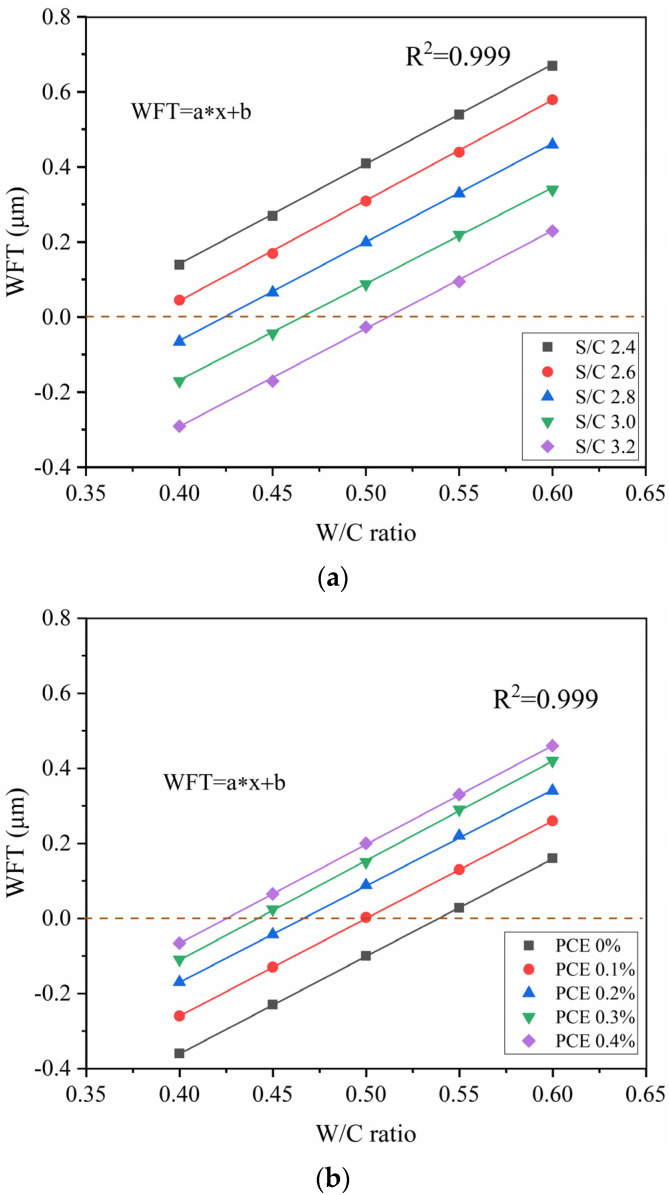
WFT versus W/C ratio: (**a**) different S/C ratios, (**b**) different PCE dosages.

**Figure 5 materials-14-04850-f005:**
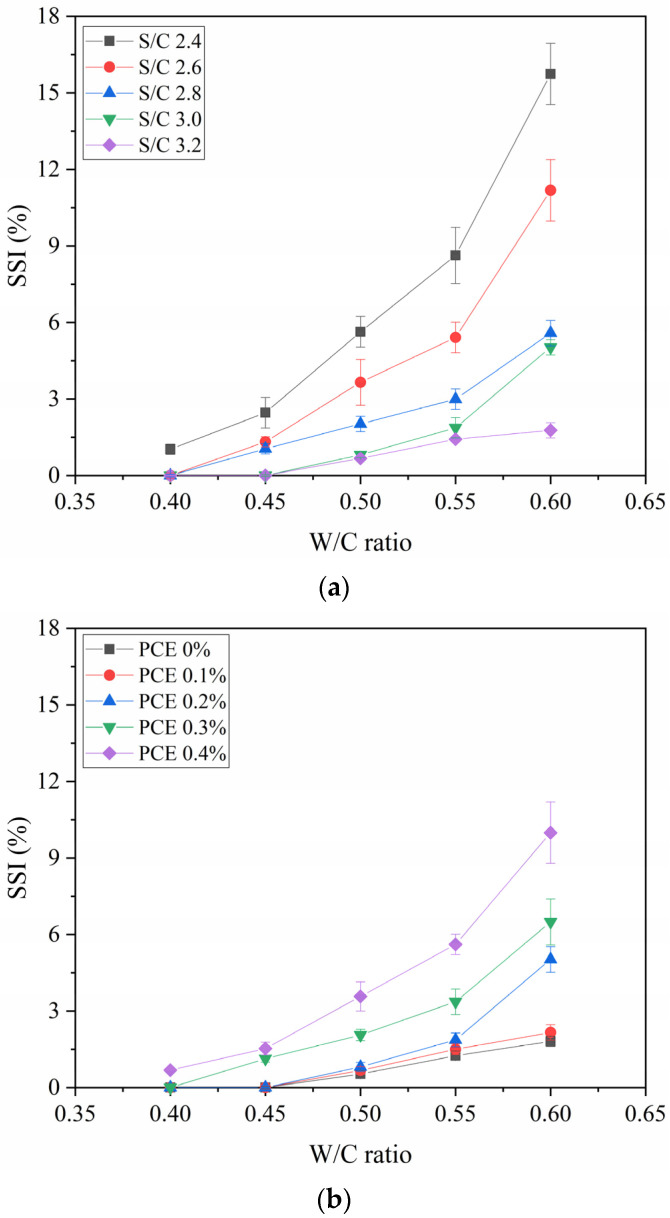
SSI versus W/C Ratio: (**a**) different S/C ratios, (**b**) different PCE dosages.

**Figure 6 materials-14-04850-f006:**
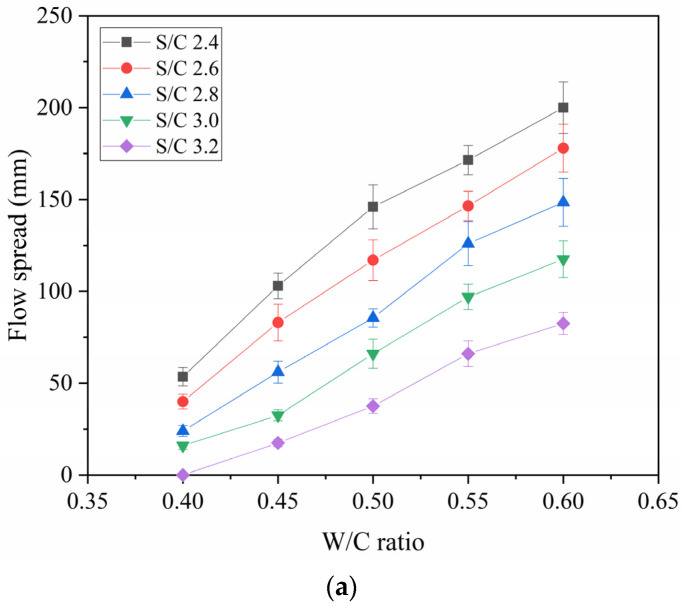
Flow Spread versus W/C Ratio: (**a**) different S/C ratios, (**b**) different PCE Dosages.

**Figure 7 materials-14-04850-f007:**
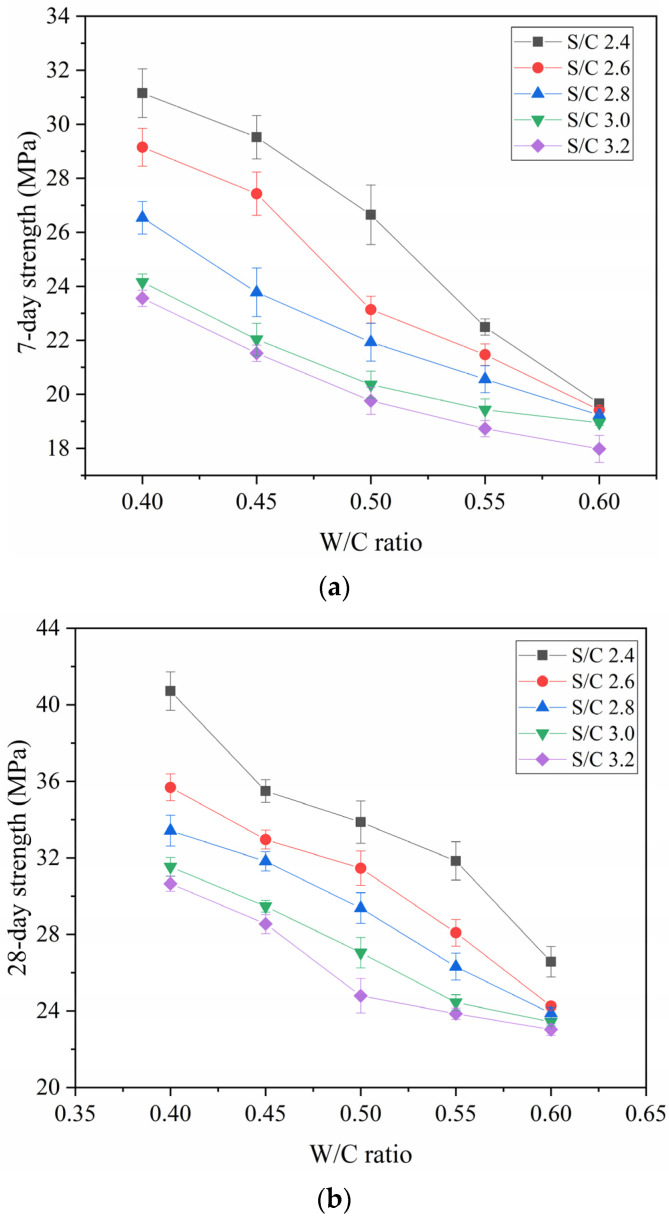
Strength versus W/C Ratio: (**a**) 7-Day Strength, (**b**) 28-Day Strength.

**Figure 8 materials-14-04850-f008:**
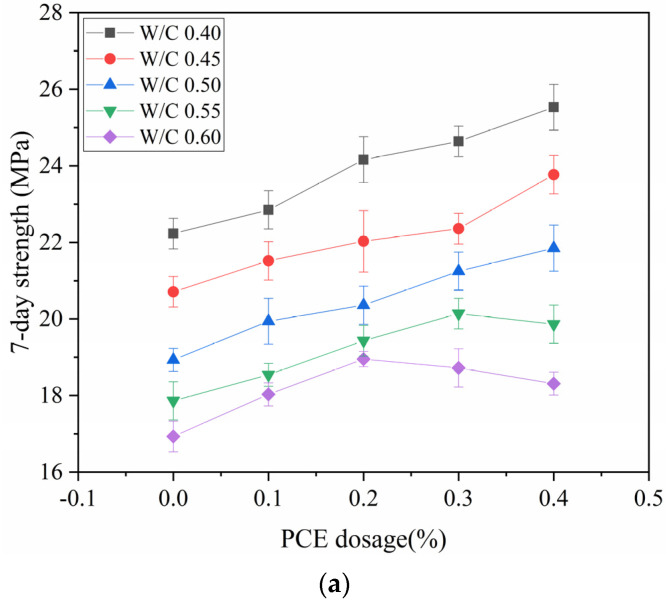
Strength versus PCE Dosage: (**a**) 7-Day Strength, (**b**) 28-Day Strength.

**Figure 9 materials-14-04850-f009:**
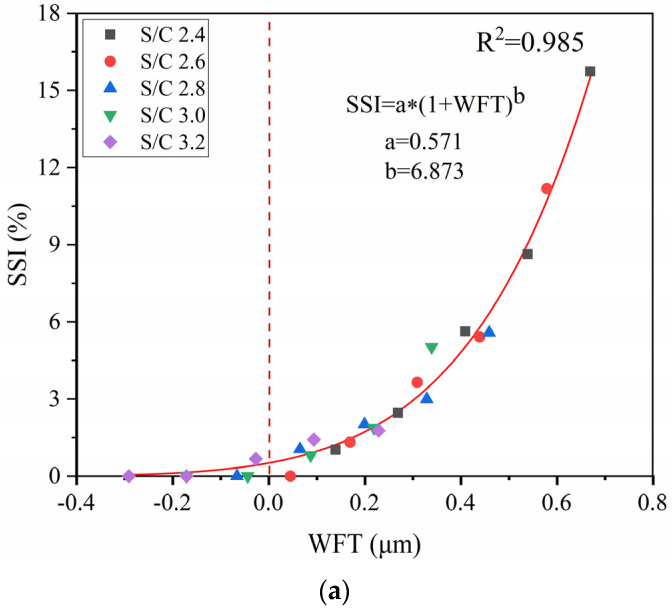
SSI versus WFT: (**a**) different S/C ratios, (**b**) different PCE dosages.

**Figure 10 materials-14-04850-f010:**
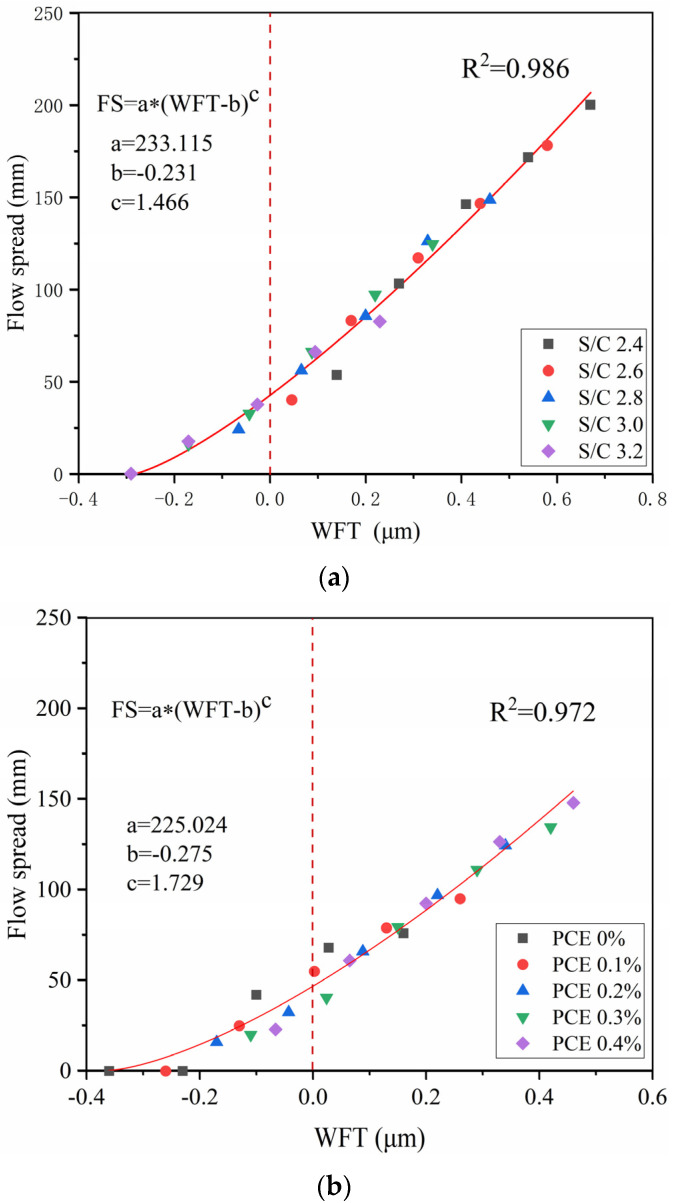
Flow Spread versus WFT: (**a**) different S/C Ratios, (**b**) different PCE Dosages.

**Table 1 materials-14-04850-t001:** Chemical compositions and physical properties of RC and fine aggregate.

Materials	Chemical Compositions/% (by Mass)							Physical Properties	
	CaO	SiO2	Al2O3	Fe2O3	MgO	SO3	LOI	Density/(g·cm^−3^)	Specific Surface Area/(m^2^·kg^−1^)
RC	63.32	20.58	5.03	3.38	2.01	2.06	1.76	3.15	347
Fine aggregate	-	>96	-	-	-	-	-	2.51	9.30

**Table 2 materials-14-04850-t002:** Mixture ratio of mortar samples.

S/C Ratio	PCE Dosage	W/C Ratio
2.4, 2.6, 2.8, 3.0, 3.2	0.2%	0.40, 0.45, 0.50, 0.55, 0.60
3.0	0%, 0.1%, 0.2%, 0.3%, 0.4%	0.40, 0.45, 0.50, 0.55, 0.60

**Table 3 materials-14-04850-t003:** Packing density and WFT results.

Mix No.	Packing Density	Water Film Thickness (μm)
2.4-0.2-0.40	0.839	0.14
2.4-0.2-0.45		0.27
2.4-0.2-0.50		0.41
2.4-0.2-0.55		0.54
2.4-0.2-0.60		0.67
2.6-0.2-0.40	0.835	0.046
2.6-0.2-0.45		0.17
2.6-0.2-0.50		0.31
2.6-0.2-0.55		0.44
2.6-0.2-0.60		0.58
2.8-0.2-0.40	0.829	−0.065
2.8-0.2-0.45		0.066
2.8-0.2-0.50		0.20
2.8-0.2-0.55		0.33
2.8-0.2-0.60		0.46
3.0-0.2-0.40	0.824	−0.17
3.0-0.2-0.45		−0.043
3.0-0.2-0.50		0.088
3.0-0.2-0.55		0.22
3.0-0.2-0.60		0.34
3.2-0.2-0.40	0.818	−0.29
3.2-0.2-0.45		−0.17
3.2-0.2-0.50		−0.026
3.2-0.2-0.55		0.095
3.2-0.2-0.60		0.23
3.0-0-0.40	0.802	−0.36
3.0-0-0.45		−0.23
3.0-0-0.50		−0.10
3.0-0-0.55		0.028
3.0-0-0.60		0.16
3.0-0.1-0.40	0.814	−0.26
3.0-0.1-0.45		−0.13
3.0-0.1-0.50		0.0026
3.0-0.1-0.55		0.13
3.0-0.1-0.60		0.26
3.0-0.3-0.40	0.832	−0.11
3.0-0.3-0.45		0.024
3.0-0.3-0.50		0.15
3.0-0.3-0.55		0.29
3.0-0.3-0.60		0.42
3.0-0.4-0.40	0.837	−0.066
3.0-0.4-0.45		0.065
3.0-0.4-0.50		0.20
3.0-0.4-0.55		0.33
3.0-0.4-0.60		0.46

**Table 4 materials-14-04850-t004:** Regressed parameters and critical value for the relationship between W/C ratio and WFT.

S/C Ratio	PCE Dosage	a	b	The Critical Value of the W/C Ratio
2.4	0.2%	2.68	−0.91	0.34
2.6	0.2%	2.70	−1.02	0.38
2.8	0.2%	2.64	−1.12	0.42
3.0	0.2%	2.59	−1.19	0.46
3.2	0.2%	2.65	−1.35	0.51
3.0	0%	2.61	−1.40	0.54
3.0	0.1%	2.61	−1.30	0.50
3.0	0.3%	2.69	−1.17	0.43
3.0	0.4%	2.65	−1.12	0.42

## Data Availability

Some or all data, models or code that support the findings of this study are available from the corresponding author upon reasonable request.

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
