# Peer review of "Influence of Sand-Cement Ratio and Polycarboxylate Superplasticizer on the Basic Properties of Mortar Based on Water Film Thickness"

_materials, 2021, doi:10.3390/ma14174850_

Round 1

Reviewer 1 Report

While there are no considerable faults in the proposed paper, the purpose and the usefulness of this research are unclear. It should be described in the Introduction section.

If someone could evaluate the fresh properties of mortar by flow spread and cohesiveness directly and easily, what is the purpose of calculating WFT using many parameters? Does WFT useful for easing daily nuisance of a concrete batching plant. How surface moisture content of sand effects on WFT?

Please correct the incomplete sentences. Examples:

  1. Under the condition of keeping the WFT positive, formulate the mixing ratio of the mortar.
  2. Calculate WFT by measuring the packing density of the mortar. So as to determine the cohesiveness and flow spread of the mortar.

Reviewer 2 Report

The reviewer has read carefully the manuscript. The reviewer would like to give comments as follows:

  1. Introduction:

Please make more clear on the novelty and purpose of this study compared with previous ones (please check the final paragraph of the introduction part carefully).

  1. Methods and materials
  • Please use the consistent words for (intermediate sand) or (fine aggregate), texts and tables are different.
  • Why the authors decided S/C ratios ranging from 2.4 to 3.2, please add comments, also please do the same for the PCE dosages.
  • Please check Equations 1 and 2 again, in addition, if you refer to any references, please make suitable citations. Regarding equation 1, the authors explain that Vb represents the total volume of cement and fine aggregate, it may not be really correct. Please check the following paper.

Kwan, A. K. H., & Fung, W. W. S. (2009). Packing density measurement and modelling of fine aggregate and mortar. Cement and Concrete Composites, 31(6), 349-357.

  • In addition, if the authors refer to any references for Equations 3 to 6, please add the souces.
  • Please add references for measuring flow spread and measuring strength.
  1. Results and discussion
  • For each set of tests, how many specimens (samples) did the authors used. It seems that the authors used 1 sample for each set. Thus, the result is not really reliable.
  • The results in Table 5 and Figs. 5 to 8 are repeated, please consider deleting Figures or Table 5?

There are many errors of typos as well as grammar in the manuscript.

Round 2

Reviewer 2 Report

Thank you for the efforts of the author to revise the manuscript. The quality of the paper is improved, however, the reviewer still has a request related to question 7.

The authors explained that 3 samples were used and the result was the average of 3 measured results. It is fine and good. Based on this, why the authors did not show the standard deviation (error bar) of results. Please kindly check it and supply it.

In addition, the way of responding to the reviewer using markup changes is not professional, this made difficulty for the reviewer. Instead of that, the authors should use highlight (using color) for the revised parts.
